# Gravity-Bench-v1: A Benchmark on Gravitational Physics Discovery for Agents

**Nolan Koblischke** [1]   **Hyunseok Jang** [1]   **Kristen Menou** [1]   **Mohamad Ali-Dib** [2]

## Abstract

Modern science emerged from reasoning over repeatedly-observed planetary motions. We present Gravity-Bench-v1, an environment-based benchmark that challenges AI agents on tasks that parallel this historical development. Gravity-Bench-v1 evaluates agents on the discovery of physics concealed within a dynamic environment, using rigorous gravitational dynamics simulations. Gravity-Bench includes out-of-distribution cases, i.e. with physics that deviates from the real world, to evaluate true scientific generalization capabilities. Agents must plan to collect data within an experimental budget and must perform a dynamic form of data analysis and reasoning to solve tasks efficiently. Our benchmark admits an open-ended space of solutions. Reference solutions for each task are provided to calibrate AI performance against human expertise. Technically at an upper-undergraduate level, our benchmark proves challenging to baseline AI agents. Gravity-Bench-v1 and planned extensions should help map out AI progress towards scientific discovery capabilities.

## 1. Introduction

The rapid evolution of artificial intelligence (AI) and machine learning has led to significant advancements in various domains, particularly in natural language processing and computer vision. However, the design of AI agents for scientific research presents unique challenges, particularly in the context of autonomously discovering new natural phenomena. Traditional benchmarks, such as those focused on knowledge evaluation (Rein et al., 2023; Hendrycks et al., 2021a; Ting et al., 2024) or general problem-solving capabilities (Clark et al., 2018; Zellers et al., 2019; Hendrycks et al., 2021b; Tian et al., 2024), fall short of what is needed

when it comes to evaluating an AI agent's capacity for discovery under normal scientific conditions of uncertainty and novelty.

To address this gap, we introduce Gravity-Bench-v1, a new benchmark specifically designed to evaluate the scientific reasoning and discovery capabilities of AI agents within a controlled, physics-based environment. This benchmark is inspired by the historical development of science (the two-body problem of gravitational dynamics) and leverages high-fidelity machine-precision simulation tools to build an environment where AI agents can interact with and explore faithful physics experiments.

In Gravity-Bench-v1, an AI agent is not merely tasked with analyzing pre-collected data but it must engage in a fuller version of the scientific process. It must schedule observations intelligently, within a constrained budget, and make inferences based on the limited and accumulating data it collects. This setup allows for an assessment of the agent's ability to reason and make autonomous decisions under dynamically-shrinking uncertainty, as observational data accumulates.

Our benchmark admits an open space of solutions, in the sense that the optimal planning for observations and algorithmic approach for quantitative answers are not a priori known. We leverage this property to offer expert solutions with uniform sampling of observations (without planning) that we consider as strong expert baselines. Our benchmark opens the possibility for an AI agent to discover planning and/or a reasoning approaches that best our expert solution, as discussed further below.

Gravity-Bench-v1 has a narrow scientific focus on the two-body gravitational dynamics problem. We are working on future expansions, including visual observation, extending to additional fields (e.g., electromagnetism), incorporating observational effects such as measurement errors, and introducing three-dimensional orbital dynamics. While various extensions in task complexity, environment realism and across physics domains are possible, we believe that the performance gap demonstrated by current AI models, even within the current limited domain highlights the immediate value of evaluation setups like Gravity-Bench-v1.

In short, Gravity-Bench-v1 challenges AI agents with

[1]University of Toronto [2]New York University Abu Dhabi. Correspondence to: Nolan Koblischke <nolan.koblischke@mail.utoronto.ca>, Kristen Menou <kristen.menou@utoronto.ca>.

*Proceedings of the $42^{nd}$ International Conference on Machine Learning*, Vancouver, Canada. PMLR 267, 2025. Copyright 2025 by the author(s).

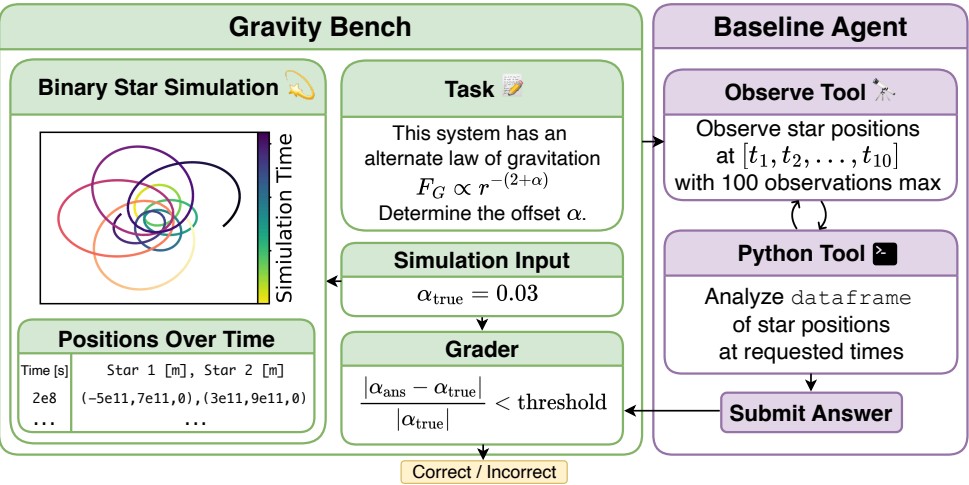

*Figure 1.* **Overview of Gravity-Bench-v1 architecture and workflow.** The binary star simulation environment (left, green) generates orbital trajectories based on input parameters, including out-of-distribution physics like modified gravity laws. An agent (right, purple) must solve physics discovery tasks by strategically collecting observations through the `observe` tool (limited to a budget of 100 observations). We evaluate against expert solutions based on full simulation data access but using uniform sampling of 100 observations as a baseline without planning. This design tests both scientific reasoning and intelligent observation planning capabilities.

tasks that mirror real-world scientific inquiry, provides a framework for evaluating their progress toward potential contributions to science, as well as their capabilities at autonomous decision-making under uncertainty. Gravity-Bench-v1 is available at https://github.com/NolanKoblischke/GravityBench and https://huggingface.co/datasets/GravityBench/GravityBench.

## 2. Related Work

Advances in leveraging AI foundation models for scientific research and discovery encompass a wide spectrum of methodologies, reflecting the diversity of tasks underpinning the scientific method (Reddy & Shojaee, 2024; Luo et al., 2025).

Specialized large language models (LLMs), such as Galactica or OpenScholar (Taylor et al., 2022; Asai et al., 2024; Sun et al., 2024), leverage domain-specific training to improve literature analysis and information retrieval. Data-driven discovery has AI systems uncover patterns in extensive datasets, typically decoupling the data acquisition from the analysis (Majumder et al., 2024; Chen et al., 2024). Automated statistical modeling focuses on deriving insights directly from existing data (Li et al., 2024). Workflow automation frameworks have AI systems propose experiments and emulate research processes (Lu et al., 2024; Siegel et al., 2024; Ma et al., 2024a; Baek et al., 2024; Ma et al., 2024b; Ghafarollahi & Buehler, 2024). Together, these methods

reflect the growing sophistication of AI foundation models in supporting, or enabling, various stages of the scientific cycle.

Many existing AI systems either treat scientific tasks in isolation, focus on specific optimizations or emphasize pattern recognition (e.g., Reddy & Shojaee, 2024; Luo et al., 2025; Yuksekgonul et al., 2024; Ma et al., 2024a). Gravity-Bench diverges somewhat by framing discovery as a dynamic, iterative process within a partially observable environment, simulating the challenges of real-world scientific inquiry. Agents in Gravity-Bench must actively explore to acquire hidden information and exploit collected data through reasoning, embodying the interplay between observation and inference that underpins natural sciences. The rigorously-simulated nature of the Gravity-Bench environment also enables evaluation in out-of-distribution scenarios, testing generalization capabilities critical for robust scientific reasoning.

Existing benchmarks have explored virtual environments for data-driven discovery or symbolic regression (e.g., Majumder et al., 2024; Jansen et al., 2024; Udrescu & Tegmark, 2020; Guimerà et al., 2020; Lemos et al., 2023; Shojaee et al., 2025), though they often focus on rediscovery of known phenomena, solving textbook-style problems or fitting equations from pre-collected datasets. By contrast, Gravity-Bench involves diverse dynamical scenarios requiring active observation scheduling, adaptive planning, and iterative scientific reasoning, mirroring the unpredictability

of real-world discovery processes. Gravity-Bench thus goes beyond curve-fitting or symbolic regression alone, as agents must proactively identify and integrate multiple quantities from strategically acquired data to solve tasks, emphasizing scientific reasoning over memorization and encouraging hypothesis formulation that is novel yet rooted in the (simulated) environment being actively explored.

Furthermore, the open-ended nature of Gravity-Bench tasks allows for diverse solution strategies, distinguishing it from tasks that emphasize solutions among preset answers. This characteristic favours exploratory reasoning and hypothesis generation in iterative cycles, rather than a more deterministic approach to measuring performance.

## 3. Benchmark design

### 3.1. Environment Design

The core design principle behind our benchmark is the concept of a rigorously-simulated, partially-observable environment.

Environments are preferred tools for evaluating agents, as they provide a dynamic setting to test capabilities, adaptability and generalization under controlled conditions. Many benchmark environments already exist in the literature, addressing a variety of domains and tasks, such as SWE-bench (Jimenez et al., 2024), RE-bench (Wijk et al., 2024), BrowserGym (Chezelles et al., 2024) or Aviary (Narayanan et al., 2024).

The engine driving our environment is a science-grade physics simulation tool. Using scientific simulation tools offers several advantages in the context of agentic benchmarks:

- Focused subdomain expertise: The simulation targets a specific subset of physics/domain knowledge on which the agent is evaluated (here: 2-body gravitational physics). Implicit knowledge (e.g. Kepler's 3rd law) can be leveraged to solve some tasks more efficiently.

- Ground truth embedding: the environment encodes ground truth in the form of input simulation parameters. They impact the environment's dynamics, which is what is observable by the agent. The agent can be thus tasked to infer the hidden ground truth or to measure/discover additional properties in more open-ended tasks, mimicking natural scientific inquiry.

- Limitless data generation: the simulation engine can generate virtually unlimited data for arbitrarily complex problems within the simulated scientific domain of interest, facilitating diverse and comprehensive evaluations.

- Modular partial observability: various observation protocols can be adopted to sparsify in time the densely simulated data, making it possible to create environments with varying levels of partial observability.

- Out-of-distribution generalization: by enabling simulation scenarios that do not occur in the real world, the engine also enables the evaluation of an agent's ability to handle novel situations and generalize beyond its regular training data, a hallmark of scientific exploration.

The observational protocol of the environment is an important design choice, effectively decoupling the densely simulated data (in time) from the sparsely observable data. Here, for simplicity, we adopt two simple observational protocols: full observability and partial observability with a finite observational budget. Within our simulated partially-observable environment,[1] planning and decision-making occurs through a dynamic form of data collection, by observational choices constrained by the environmental protocol (see Figure 1 for an overview of the benchmark). In a follow-up work, we also consider the addition of vision as an observational modality of the environment, enabling the evaluation of visual perception and reasoning as additional agent capabilities.

All simulations are implemented using `Rebound` (Rein & Liu, 2012; Tamayo et al., 2020), the current gold standard for gravitational few-body dynamics. Our standard `Rebound` simulation takes as input the stellar binary parameters (point masses, 3D positions, and 3D momentums), and any additional forces present. `Rebound` then solves Newton's gravity equations forward in time, for mostly 10 orbits. We save only the stars' Cartesian positions as a function of time for the agent to access in solving the problem.

### 3.1.1. OBSERVATION PROTOCOL AND TOOL

In this version of Gravity-Bench, all orbits are in the (x,y) Cartesian plane by construction (i.e. z=0 at all times). This is closely related to the ideal 'face-on' geometry of real observations, where a binary's orbital plane coincides with the plane of the sky. Geometric projection effects, which are paramount in more realistic observations, will be addressed in a subsequent benchmark version.

In Gravity-Bench-v1, we adopt two environment observa-

---

[1] Here, partial observability refers to the agent having only access to time-selected snapshots from the densely simulated data available in the environment. Even though our setup promotes planning under uncertainty and active information acquisition, concepts found in the POMDP literature (e.g., Kaelbling et al., 1998; Bowyer, 2021), we note that our observation protocol assumes error-free observations. This idealization can be relaxed in future benchmark iterations to additionally make the environment stochastic.

tion protocols, which are mediated by an observation tool made available to the agent. In the first 'full-obs' protocol, the agent has access to the full dense set of simulation data.

In the second 'budget-obs', the agent is permitted a maximum pre-determined number of observations, $N_{\text{obs}}$, constrained to be within the time range covered by the dense simulation data. The agent is free to choose which times to observe,[2] for up to 10 data points per observation-tool call, subject to a maximum total of $N_{\text{obs}}$. In the 'budget-obs' protocol, the agent is therefore incentivized to be strategic within the observational budget allocated and to reason along as more observations are collected in several modular steps.

In practice, the agent repeatedly queries the observation tool with a series of observation times, and the tool returns the corresponding data, appending them to previously-collected observations.[3]

### 3.2. Scientific problems

Binary star systems modeled as point masses offer a rich enough abstraction for our first benchmark to cover a wide range of potential problems. In particular, this includes tasks that involve inferring hidden physical properties from limited observational data. For some tasks, the target values are direct parameter inputs into the simulation such as component masses. Other tasks involve finding values that are not direct inputs but can be derived from the simulation data, such as a star's average distance from the center of mass, the fraction of time acceleration is below the mean, or the time it takes a star to travel 20% of its orbital path.

We first design a diverse set of two-body gravitational simulations, illustrated in Figure 2. We deliberately incorporate symmetry-breaking strategies, such as displacing the center of mass from the system origin or introducing uniform center-of-mass drift (known as "proper motion"), to mirror the messy realities encountered in genuine astronomical observations.

We then design tasks to be solvable only through careful derivation requiring success at multiple intermediate steps. This aligns with the scientific process in reality. For example, to determine the total energy of the system, one must first find both stellar masses, which in turn require estimates of accelerations and separations.

---

[2]Cubic interpolation between densely simulated timesteps provides near continuous-time coverage with minimal numerical errors.

[3]The agent is allowed to observe at any point back in simulated time. An observation protocol that would more closely match observations in the real world would constrain any new observations to occur only forward in time relative to previously-collected observations. We will explore additional observation protocols in future work.

In addition to standard Newtonian gravity, we introduce six scenarios that deviate from real-world physics. Three incorporate a drag force, requiring agents to infer the drag timescale from shrinking orbits, and three adopt a modified gravitational exponent with a force of gravity $F_G \propto r^{-(2+\alpha)}$, where $r$ is the separation between stars, and the task is to determine $\alpha$ (which is 0 in Newtonian gravity). We emphasize that our drag and modified gravity laws represent scenarios rarely, if ever, considered in standard textbooks or physics literature on two-body dynamics, hence they are out-of-distribution within the context of our benchmark and likely require compositional generalization from physics learned in other contexts.

From our 16 two-body simulations, we design 50 tasks, 47 with numeric answers and 3 true/false. We match each task with multiple simulation variations for a total of 206 task-simulation pairs.

## 4. Experiments

### 4.1. Evaluation details

For each task, we implement an expert solution based on only the data available to the AI agent and confirm the solutions agree with the simulation inputs if directly available for that task or a superior solution based on additional information from the `Rebound` simulation (such as a built-in evaluator for orbital parameters).

These expert-reference solutions were produced by one PhD student and one research scientist with domain expertise. Each of the two independently double-checked every answer, and the final values were cross-verified against `Rebound`'s built-in orbital diagnostics whenever available.

Under the constraint of an observation budget, the performance of our expert algorithmic solutions depends on the observation strategy. Therefore, we use the performance of a baseline strategy to set task-specific error thresholds that an AI answer must beat to be marked correct. Specifically, we define our baseline expert reference (expert-ref-$N_{obs}$) as the performance obtained using $N_{obs}$ observations equally spaced in time (without planning), since optimal observation strategies for all simulation-task pairs are costly to develop. When evaluating AI agents with an observation budget of $N_{obs} = 100$ (budget-obs-100), we evaluate the AI agent performance against this 'expert-ref-100' baseline.

An answer is marked correct if its percentage error relative to our ground truth answers falls at or below the task-specific maximum permissible threshold. These thresholds account for the inherent difficulty of solving each task with limited observations. They are set based on the performance gap between our expert solution using full simulation data (expert-ref) versus 100 uniform observations (expert-ref-

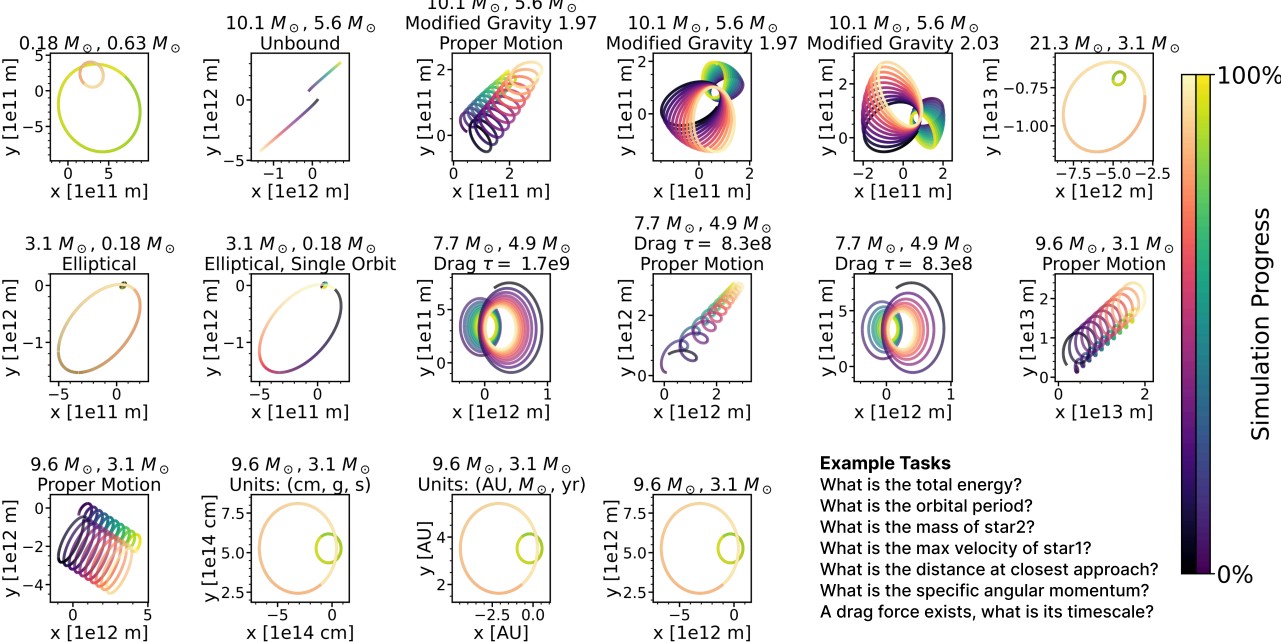

Figure 2. **Overview of the gravitational simulations used in the benchmark.** Each panel shows the orbital trajectories of a binary star system in the x-y plane, with masses indicated in solar masses ($M_\odot$). The color gradient indicates simulation progress from start (dark) to end (light). Simulations include standard orbits, systems that are unbound, systems with modified gravity, systems with drag forces, systems with proper motion, etc. The benchmark also includes versions of the same system represented in different units, to evaluate unit handling. Sample questions that could be asked about these systems are shown.

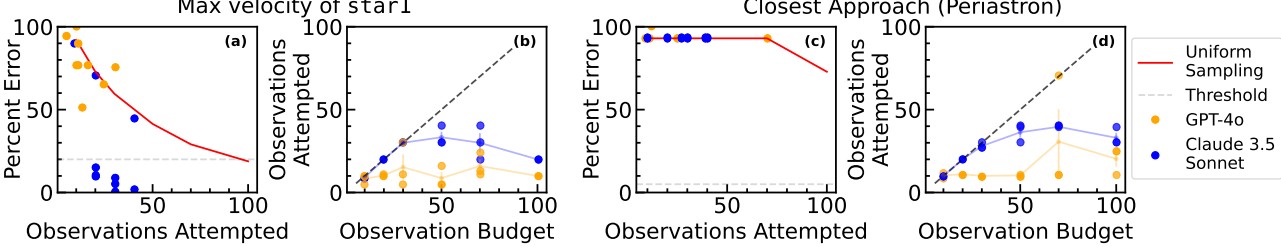

Figure 3. **Agent performance in finding the maximum velocity of a star under various observational budgets.** *(a)* Percent error for each agent as a function of the total number of observations used, where each point represents an individual run. Uniformly sampling in time with a expert solution (red line) serves as a planning-free baseline. Claude 3.5 Sonnet (blue) sometimes refines its observations enough to achieve under 1% error, while GPT-4o (orange) shows less consistent improvement. *(b)* Observations attempted by each agent as a function of the max allocated observation budget. Points show individual runs, while lines with error bars show the mean and standard error across runs for that budget. While an ideal approach would exploit all available observations (dashed line), both GPT-4o and Claude 3.5 Sonnet stop early, often using fewer than half of the available observations for budgets above 30. This underutilization highlights a lack of robust planning and answer verification. *(c), (d)* Percent error in finding the periastron distance in a single, highly elliptical orbit where the stars spend only 0.2% of the time within 5% of the closest approach. As discussed in Appendix D, an expertly planned solution can achieve 2% error with 50 observations, but our uniform-sampling baseline with 100 observations (without planning) performs poorly (70% error), as do both AI agents. The horizontal dashed lines indicate the threshold by which the agents are marked in budget-obs-100.

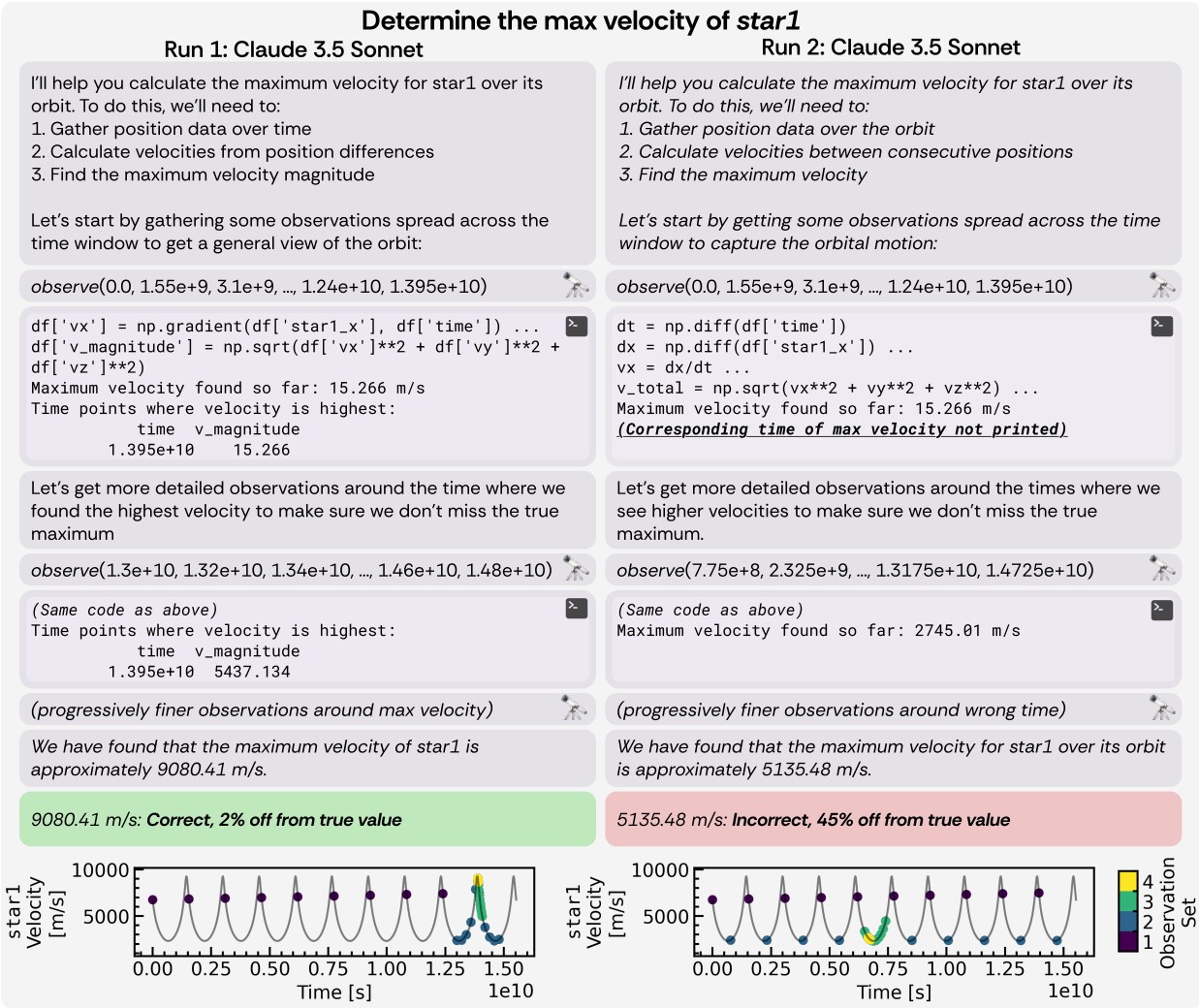

*Figure 4.* **Two observation-planning runs by Claude 3.5 Sonnet on the same task using 40 observations.** The figure highlights how minor differences in planning lead to drastically different outcomes. In each run, the agent collects position data in multiple steps, computes velocity from finite differences, and refines its search for the peak velocity. *Top panels:* Excerpts of the agent's traces including the planning, observations, and code use. On the left, the agent systematically tracks the highest velocity times, and progressively refines its estimate, achieving a final error of only 2%. On the right, however, the agent never accurately records peak-velocity times, and proceeds to query intervals around low velocity times, resulting in a 45% error. It seems to misinterpret increasing velocity estimates from finer time resolution as evidence of higher true velocities, rather than as improved measurement accuracy. *Bottom:* True velocity curves (gray) overlaid with the agent's observations (colored dots). Later queries appear in brighter hues, showing how an intelligently planned approach can converge near to correct velocity, while a misplanned approach (right) fails to capture the velocity peak.

*Table 1.* **Model performance.** The *Performance* column shows the percentage of tasks each model solves within a question-specific error threshold. Under *budget-obs-100*, a maximum of 100 observations can be requested, testing the ability to plan. The model is marked correct if it is within the threshold determined for each task (between 5% and 70%, see Section 4.1). Under *full-obs*, each model can access all observations and is marked correct if within 5% of the correct answer. We also report total cost, total run time, and average observations used. Results averaged over three runs with standard errors shown.

| | Score | Total Cost ($) | Total Time (min) | Mean Observations Used |
|---|---|---|---|---|
| **Sequential Observations - 100 Observation Budget** | | | | |
| o4-mini-high-2025-04-16 | **49.4%** $\pm$ **2.6%** | 81.23 $\pm$ 1.68 | 2854.5 $\pm$ 151.7 | 33.2 $\pm$ 1.5 |
| claude-3-5-sonnet-20241022 | 21.5% $\pm$ 2.5% | 15.88 $\pm$ 0.64 | 128.3 $\pm$ 2.5 | 24.3 $\pm$ 0.5 |
| claude-3-5-haiku-20241022 | 16.1% $\pm$ 2.3% | 3.33 $\pm$ 0.10 | 94.4 $\pm$ 0.7 | 12.6 $\pm$ 0.4 |
| gpt-4o-2024-11-20 | 15.5% $\pm$ 2.1% | 9.60 $\pm$ 0.12 | 63.5 $\pm$ 1.6 | 12.2 $\pm$ 0.7 |
| gpt-4o-mini-2024-07-18 | 8.3% $\pm$ 1.5% | 0.60 $\pm$ 0.03 | 83.3 $\pm$ 4.2 | 13.4 $\pm$ 1.0 |
| **Full Table Access** | | | | |
| o4-mini-high-2025-04-16 | **73.9%** $\pm$ **2.4%** | 15.59 $\pm$ 0.24 | 522.6 $\pm$ 14.2 | - |
| claude-3-5-sonnet-20241022 | 39.5% $\pm$ 3.2% | 5.58 $\pm$ 0.06 | 69.9 $\pm$ 0.3 | - |
| gpt-4o-2024-11-20 | 36.1% $\pm$ 3.2% | 3.41 $\pm$ 0.17 | 45.6 $\pm$ 3.2 | - |
| claude-3-5-haiku-20241022 | 34.1% $\pm$ 3.1% | 1.46 $\pm$ 0.03 | 63.3 $\pm$ 1.3 | - |
| gpt-4o-mini-2024-07-18 | 26.7% $\pm$ 2.8% | 0.16 $\pm$ 0.00 | 40.4 $\pm$ 0.7 | - |

100): $\frac{|\text{expert\_ref}(100) - \text{expert\_ref}(\text{full-obs})|}{\text{expert\_ref}(\text{full-obs})}$. Tasks where uniform sampling achieves near-full-data performance (e.g., orbital period estimation) receive strict thresholds (5%), while those where 100 uniform observations are insufficient (e.g., maximum velocity measurement) allow larger margins (20%). For extreme cases like measuring the exponent of a modified gravitational force, where expert-ref-100 shows >1000% error, we set lenient but achievable thresholds (70%). To show this is achievable for this task, we find that an expert solution can reach within 1.7% error of the ground truth gravitational exponent with 70 elaborately planned observations (see Appendix C). For most problems, only the combination of a strong algorithmic solution and observational strategy leads to a high quality answer.

We also design our tasks to resist random guesswork. In the modified gravity case, the model must find the deviation from Newtonian gravity ($r^{2+\alpha}$) rather than the full exponent ($r^{\alpha}$), as 0.03 is much harder to estimate within 70% (correct range: $\alpha \in [0.009, 0.051]$) than 2.03 for example.

### 4.2. Baseline agent

Our observation protocol (Section 3.1.1) requires planning future observations based on existing observations, making single-step solutions unlikely to succeed. Therefore, our benchmark is designed to evaluate AI systems that operate as agents that probe the environment and perform actions over multiple steps. We design a baseline agent around a ReAct-style scaffold (Yao et al., 2023). The agent can use our `observe` tool and a Python interpreter adapted from Langchain (Chase, 2022) with access to packages like `numpy` (Harris et al., 2020), `scipy` (Virtanen et al., 2020) and `pandas` (pandas development team, 2020) and receive outputs and exception tracebacks. While we evaluate this specific configuration, our benchmark supports arbitrary

agent architectures. The baseline agent prompts are shown in Appendix F.

### 4.3. Performance of baseline agent

We test OpenAI (OpenAI, 2024; 2025) and Anthropic (Anthropic, 2024) models in Table 1[4]. o4-mini-high achieves the highest performance among tested models with Claude 3.5 Sonnet in second. When constrained by a 100-observation budget, each model shows a significant performance drop, suggesting observational planning remains challenging. The *full-obs* evaluation configuration requires less computation time than *budget-obs-100* because all observations are immediately available to the agent.

Impressively, o4-mini-high consistently solves multiple runs across all modified gravity tasks in *full-obs*, accurately estimating the exponent in scenarios with gravitational forces following $F_G \propto r^{2+\alpha}$. This specific modification of gravity is rarely discussed in textbooks or literature in the two-body dynamics context. This suggests generalization to novel physical scenarios, although more evaluations in this regime are warranted. In *budget-obs*, it fails to consistently solve the out-of-distribution problems.

The only other models to solve some OOD tasks were Claude 3.5 Sonnet, which consistently solves one modified gravity task and GPT-4o mini, which unexpectedly solves a single modified gravity task only once out of three runs. This might demonstrate the power of repeated sampling even of less capable models, as discussed in Brown et al. (2024).

---

[4]We note that the Claude 4 models were released during the final stages of this work and thus was not included in our analysis (Anthropic, 2025).

## 4.4. Planning

An elaborately planned observational strategy is required to solve problems efficiently under the restriction of a budget. This is most evident in problems such as finding the maximum velocity of a star since velocity is highest only over the small fraction of an elliptical orbit when the two stars are closest to eachother. Uniformly sampling observations in time is not sufficient to determine this max velocity, as evidenced by the performance of our expert-ref-100 solution ($\sim$20% off).

To investigate planning ability, we provide the AI agents tasks, including finding the maximum velocity of a star and finding the minimum separation of the two stars (the periastron), but vary the allowable max budget from 10 to 100 observations. Figure 3 summarizes these runs by plotting the agent's error against the number of observations the agent decided to conduct. Elaborate planning is required to significantly outperform the expert-ref-100 baseline. This baseline achieves very high error ($> 90\%$) with 10 observations to moderate error ($20\%$) with 100 observations, reflecting the difficulty of pinpointing the specific max-velocity region with uniform temporal sampling (the minimum separation task shows similar results as discussed in Appendix D).

A highly effective strategy would use a substantial fraction of the budget to iteratively refine estimates around the time ranges suspected of containing the maximum velocity. However, we observe that all models tested request very few observations relative to their budgets. For budget-obs-100, GPT-4o only uses 12 observations on average (Table 1) while Claude 3.5 Sonnet uses twice as many. This might explain Claude's higher performance but is still a vast underutilization of the budget. Nonetheless, the experiment in Figure 3 shows that Claude 3.5 Sonnet solves this max velocity task to within 1% due to elaborately observation planning.

We find that the agents stop further observation once a plausible solution is found rather than observing further to confirm their answer (e.g. via denser sampling or re-checking of other orbital regions). These experiments indicate that the tested models tend to prematurely rush to solutions and systematically underutilize their observational budget when working in the baseline agent framework, an issue we have not yet fully explained and explicitly identify as an important direction for future automated analysis of agent traces, given the extensive length and complexity of these traces.

## 4.5. A case study on planning

Figure 4 presents two runs by Claude 3.5 Sonnet, finding the maximum velocity of a star with 40 observations. The left run demonstrates a more effective strategy. It begins with a broad temporal sampling to identify regions with higher velocity, though these initial velocity estimates are imprecise due to the coarse time resolution. Upon detecting a high-velocity region at a specific time, the agent implements progressively finer temporal resolution around this interval. This iterative refinement approach achieves a final velocity measurement within 2% of the ground truth. Notably, the agent maintains a record of the highest velocity magnitudes and corresponding times, enabling targeted subsequent observations in regions of interest.

In contrast, the run in the right panel fails to converge on the correct velocity. While attempting a similar strategy, the agent neglects to track the times of peak velocities. When the time resolution is refined, the velocity estimates appear to increase, but this reflects improved measurement accuracy rather than finding a truly higher-velocity region. Consequently, subsequent observations are mistakenly concentrated in regions of lower velocity, and the agent reports a velocity that deviates from the true maximum by approximately 45%.

## 4.6. Failure modes

We observe that the models incorrectly assume symmetry in the system. For example, we observe that they often wrongly assume the center of mass is at the origin (0, 0, 0), or they neglect that the system can have drift (see simulations labeled "proper motion" in Figure 2) when finding orbital properties that are critical to solving the problems, leading to incorrect answers.

Our findings reveal a tendency for AI models to take shortcuts rather than systematically derive intermediate quantities. For example, they frequently bypass calculating the mass of the stars directly and instead assume a value, such as 1 gram, to continue with the problem. As detailed in Appendix E, we find that solutions containing such mass assumptions correlate strongly with incorrect answers across all models tested. Notably, GPT-4o makes a mass assumption in 33% of incorrect solutions compared to 5% of correct solutions. Even Claude 3.5 Sonnet, which performs better overall, shows more than double the rate of mass assumptions in incorrect responses versus correct ones.

## 5. Conclusion and Discussion

We introduced Gravity-Bench-v1, a novel benchmark designed to evaluate AI agents in tasks that emulate the scientific discovery process, requiring iterative reasoning, dynamic planning, and robust generalization. By challenging agents with both standard and out-of-distribution physics scenarios, the benchmark aims to assess scientific reasoning capabilities beyond memorized solutions. Our results demonstrate that while baseline AI models perform moderately well with the full table of observations, they struggle

under constrained observational budgets, often failing to plan or exploit available data effectively. These findings highlight current limitations in long-horizon reasoning and adaptive decision-making, which are important components for autonomous scientific discovery.

Looking ahead, Gravity-Bench-like approaches have significant potential for growth as tools for advancing AI research in scientific reasoning. By expanding the benchmark to include incrementally more complex physics, one can aim to map out progress toward AI systems capable of genuine contributions to science. Additionally, this type of benchmarks with controlled environment and open-ended solution space may provide opportunities to characterize the robustness of autonomous AI agents in handling novel and uncertain scenarios, an issue connected to safety. Finally, adapting environments like Gravity-Bench-v1 for reinforcement learning has the potential to serve as a stepping stone towards building AI agentic systems that not only analyze but also explore and innovate in the domain of scientific discovery.

## Acknowledgements

Work in the AI Physics and Safety Lab at the University of Toronto is supported by the National Science and Engineering Research Council of Canada, the Dunlap Institute of Astronomy & Astrophysics (seed funding), Open Philanthropy, Anthropic (compute credits) and OpenAI (superalignment grant). MAD was supported by Tamkeen under the NYU Abu Dhabi Research Institute grant CASS.

We would furthermore like to acknowledge that this work was performed on land which for thousands of years has been the traditional land of the Huron-Wendat, the Seneca and the Mississaugas of the Credit. Today this meeting place is still the home to many indigenous people from across Turtle Island and we are grateful to have the opportunity to work on this land.

## Author Contributions

KM proposed the simulated observable environment approach adopted in GBv1, the various task categories, guided the benchmark design throughout and wrote several sections. NK designed and wrote most of the benchmark code-base, specific tasks, and implemented all expert problem solutions. NK also wrote the bulk of the paper content and produced all figures. HJ contributed substantially to the code-base. MAD provided domain-knowledge validation on all the code, tasks and solutions, designed the table detailing all tasks and wrote several sections.

## Impact Statement

This paper presents work whose goal is to advance the field of Machine Learning. There are many potential societal consequences of our work, none which we feel must be specifically highlighted here.

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

## A. `Rebound` simulations details

All simulations are implemented using `Rebound`, a popular gravitational N-body integrator. The core of this framework is the ordinary differential equation formulation and the numerical time-integrator, to achieve machine precision and minimize error build-up over time. For most problems we use `WHFast` (Rein & Tamayo, 2015), an unbiased, machine precision, energy conserving integrator. The integration timestep is conservatively chosen to be one-five-thousandth of the system's orbital period. For problems where forces other than gravity are present, or the gravitational law has been modified, `WHFast` is not adequate. We then use `IAS15`, an adaptive time-step 15th-order integrator where errors are kept to below machine precision.

Our standard `Rebound` simulation takes as input the stellar binary parameters (point masses, 3D positions, and 3D momentums), in addition to the integrator choice discussed above, and any additional forces present. `Rebound` then solves Newton's gravity equations forward in time, for 10 orbits[5]. At densely-sampled timesteps, it outputs the complete Cartesian and orbital elements for both stars. This detailed information is saved for reference as part of the environment but is not provided to the agent. Instead, we separately save only the stars' Cartesian positions as a function of time for the agent to access in solving the problem, modulo the observation protocol.

## B. Description of the benchmark problems

| Question | Solution | Comments |
|---|---|---|
| mass_largest_star
Determine the mass of the largest star. | Calculate the separation magnitude of the two stars using their 3D positions. Compute the acceleration of star2 by taking the second derivative of its position with respect to time. Using the acceleration and separation magnitudes $a$ and $d$, calculate the mass of star1: $M_1 = \frac{a_{\text{star2}} \cdot d^2}{G}$ Repeat the process for star1 to calculate the mass of star2. Finally, return the maximum of the two masses: largest_mass = $\max(M_1, M_2)$. | The separation magnitude array is given by $d = \sqrt{(x_1 - x_2)^2 + (y_1 - y_2)^2 + (z_1 - z_2)^2}$. The gravitational acceleration is given by $a = \frac{GM}{d^2}$ as the force is $F_1 = \frac{GM_1 M_2}{d^2} = M_1 a$. This illustrates how stellar masses are determined whenever needed, for all the rest of the problems. |
| max_velocity_star1
Calculate the maximum value of velocity for star1 over the orbit. | Calculate the velocity magnitude array of star1 by computing the first order time derivative of its 3D position. Return the maximum of the array. | The velocity magnitude array is given by $V_1 = \sqrt{(v_{x1})^2 + (v_{y1})^2 + (v_{z1})^2}$ where $v_{x1} = dx_1/dt$. Final Answer = MAX($V_1$). This illustrates how stellar velocities are determined whenever needed, for all the rest of the problems. |
| max_acceleration_star1
Calculate the maximum value of acceleration for star1. | Take the second order derivative of position with respect to time to get the acceleration array then calculate its magnitude. Return the maximum of the acceleration magnitude array. | $a_1 = \sqrt{(a_{x1})^2 + (a_{y1})^2 + (a_{z1})^2}$ where $a_{x1} = d^2 x_1/dt^2$. This illustrates how stellar accelerations are determined whenever needed, for all the rest of the problems. |

---

[5]Except for `Linear_drag` and `Modified_gravity`

| | | |
|---|---|---|
| semi_major_axis
Determine the semi-major axis of the system's orbit. | Calculate the total semi-major axis by finding the magnitude of the 3D stellar separations and averaging their extremes (min/-max).
For verification, set up a Rebound Simulation. Add the two stars to the simulation, specifying their masses, positions, and velocities. Compute the orbital elements for star2, with star1 as the primary. Extract and return the semi-major axis from the orbital parameters. | The total semimajor axis $a$ is given by $a = \frac{\max(d) + \min(d)}{2}$ where $d$ is the separation array. For other questions, you can calculate individual semi-major axes by weighting the total axis by each star's mass ratio to the total system mass. This illustrates how stellar semimajor axis are determined whenever needed, for all the rest of the problems. |
| eccentricity
Determine the eccentricity of the system's orbit. | Calculate an array of stellar separation magnitudes along with its maximum and minimum values ($r_{max}$ and $r_{min}$). The eccentricity is given by ($r_{max}$ - $r_{min}$) / ($r_{max}$ + $r_{min}$). For verification, initialize a Rebound simulation, add the stars, and obtain the eccentricity from the simulation. | Orbital eccentricity is a dimensionless parameter that measures the deviation of an orbit from being perfectly circular. This illustrates how stellar eccentricities are determined whenever needed, for all the rest of the problems. |
| semi_minor_axis
Determine the total semi-minor axis of the system's orbit. | Calculate the total semimajor axis $a$ and the eccentricity $e$. The semiminor axis $b$ is given as $b = a \cdot \sqrt{1 - e^2}$. For verification, set up a Rebound Simulation. Add the two stars to the simulation, specifying their masses, positions, and velocities. Compute the orbital elements for star2, with star1 as the primary. Extract semi-major axis and eccentricity from the orbital parameters. Calculate and return the semi-minor axis. | This illustrates how stellar semiminor axis are determined whenever needed, for all the rest of the problems. |
| period
Determine the orbital period of the system. | Calculate the magnitude of stellar separations using their 3D positions. Use scipy.signal.find_peaks to identify the peaks (maximums) in the separation array, which correspond to the apoastron points. From these peaks, calculate the average period by taking the mean difference between consecutive peak times. Return this value. | Apoastron is the point in the orbit of a star or other object in a binary system where it is farthest from its companion. This illustrates how stellar periods are determined whenever needed, for all the rest of the problems. |

| | | |
|---|---|---|
| time_fraction_acceleration _below_mean Calculate the fraction of time in a single orbit where the acceleration of star1 is below the mean acceleration. | Calculate the orbital period. Find the index corresponding to a time just after one full period and use this to locate the time of the next pericenter passage. Isolate the data for a single orbit by selecting the rows where the time is between the pericenter passage and one full period later. Calculate the mean acceleration of star1 for this orbit. Identify the time intervals where the acceleration of star1 is below the mean. To account for potential irregularities in time steps, sum the time differences where the acceleration is below the mean. Finally, compute the fraction of the total orbital period during which the acceleration is below the mean and return this value. | - |
| 2K+U Determine the quantitative value of 2K + U for the system in joules. | Calculate the stars' masses, velocities and separation magnitudes to compute the system's instantaneous kinetic (K) and potential (U) energies. Average the kinetic and potential energies arrays, then return 2K+U. | From the Virial theorem we have 2K + U = 0, where K is the total kinetic energy and U the gravitational potential energy. This should be satisfied close the machine precision in the simulated environment. |
| apoastron Determine the apoastron of the system's orbit. | Calculate the stellar semimajor axis and eccentricities. The apoastron is defined as $a(1 + e)$. For verification, initialize a Rebound simulation and set its units. Add both stars to the simulation with their masses, positions, and velocities. Compute the orbital parameters of one star relative to the other using Rebound. Finally, return the apoastron, the maximum separation along the orbit. | - |
| area_swept_over_time_apo Calculate, at apoastron, the rate of area swept per unit time by the imaginary line joining star1 to star2. | Calculate relative positions and velocities between the two stars using finite differences then compute the specific angular momentum vector and half its magnitude. For verification, initialize a Rebound simulation, add the stars, and obtain the specific angular momentum from the simulation. | Kepler's 2nd law: the rate of area swept is half the specific angular momentum of the system given by $h = \sqrt{h_x^2 + h_y^2 + h_z^2}$ with $h_x = y \cdot v_z - z \cdot v_y$, where (x,y) is the orbital plane. |
| avg_distance_COM_star1 Calculate the time-averaged distance between star1 and the COM over a single orbit. | Calculate the stellar masses along with the orbital period and the time of pericenter passage. Isolate a single orbit by filtering the data between the pericenter passage and one period later. Calculate the distance between star1 and the COM during this orbit. Finally, integrate the distance over time and divide by total time to get the time-averaged distance. Return this value. | COM: center of mass. The distance between star1 and the COM is given by $d_{\text{star1}} = \sqrt{(x_{\text{star1}} - x_{\text{COM}})^2 + (y_{\text{star1}} - y_{\text{COM}})^2 + ...}$, with $x_{\text{COM}} = \frac{m_1 \cdot x_{\text{star1}} + m_2 \cdot x_{\text{star2}}}{m_1 + m_2}$. Analogous equations can be written for $y_{\text{COM}}$ and $z_{\text{COM}}$. |

| is_bound Determine whether the system is gravitationally bound (True) or unbound (False). | Compute the velocity magnitudes for both stars. Calculate the gravitational potential energy (U) using the separation between the stars and their masses. Calculate the kinetic energy (K) using the squared velocities and masses. Compute the mean kinetic and potential energies over the dataset. Check whether the total energy (K + U) indicates a bound system. Return True if the system is bound. | Unbound system: K + U > 0, adopting the usual notation of zero potential energy at infinite separation. |
|---|---|---|
| kepler_third_law Determine if Kepler's third law is satisfied. Answer: True if Kepler's third law is satisfied, and Answer: False if it is not. | Calculate the stellar masses, periods, and semimajor axis. Compute the ratio $P^2/a^3$ from the data and compare it to the theoretical value: $\frac{4\pi^2}{G(m_1+m_2)}$. If the percentage difference is less than 0.1%, return True. | Kepler's Third Law: the squares of the orbital periods of the planets are directly proportional to the cubes of the semi-major axes of their orbits. |
| linear_drag This system experiences a drag given by $acc_i = -v_i/\tau\ \hat{\imath}$ for the i-direction, where $acc$ is the acceleration, $v$ is the velocity, and $\tau$ a timescale. Calculate the value of the coefficient of linear drag, $\tau$, for the system. | Calculate the separation between the stars using their 3D coordinates. Identify the peaks (apoastron) and troughs (periastron) in the stellar separation data using scipy.signal.find_peaks. Ensure both arrays (peaks and troughs) have the same length by truncating the longer one if necessary. Compute the semi-major axis for each orbit by averaging the apoastron and periastron distances. Find the average time for each orbit by averaging the times corresponding to the peaks and troughs. Define an exponential decay function to model the semi-major axis as a function of time: $a(t) = a_0 \times exp(-2t/\tau)$. Use scipy.optimize.curve_fit to fit this model to the semi-major axis data and extract $\tau$. | - |
| max_angular_velocity_star1 Calculate the maximum value of angular velocity for star1 over the orbit. | Calculate the velocity vector **v** of star1 and the magnitude of the 3D stellar separation $r$ . The angular velocity vector of star1 is then given by $\omega = \frac{\mathbf{r_1}\times\mathbf{v_1}}{r^2}$. Finally, return the maximum of the resulting array. | This can be retrieved for verification from Rebound using the specific angular momentum $h$: $\omega = \frac{h}{r^2}$ |
| max_momentum_star1 Calculate the maximum linear momentum for star1 over the orbit. | Calculate the mass and the velocity magnitude of star1. Calculate the linear momentum array as the mass × velocity. Return the maximum of the array. | - |
| modified_gravity_power_law This system is governed by an alternative law of gravitation where the r dependence is $r^{-(2+\alpha)}$ where alpha represents the deviation from Newton's inverse square law. Calculate $\alpha$. | Calculate the stellar separation magnitudes and the total acceleration of one of the stars. Take the logarithm of both separation (r) and acceleration (a). Fit a linear function to log(a) vs log(r), but only for: Separations above the median separation and data points without outliers (using MAD-based outlier removal). The slope of this fit gives us -(2 + $\alpha$), so $\alpha = |\text{slope}| - 2$ | Out-of-distribution example where the orbits are effectively precessing. In reality $F_G \propto r^{-2}$, and thus other exponent values are unlikely to be present in pre-training data. |

| | | |
|---|---|---|
| multiply_mass_period
Determine the factor X by which the central mass should be multipled for the orbital period of the system to be 21 days. You can assume the central mass is star1 which is much more massive than star2. | Compute the average period $P$ of the system. Calculate and return the factor $X$ using Kepler's Third Law: $X = \left(\frac{P}{21}\right)^2$ | - |
| orbital_area
Determine the total area encompassed by the orbit of the system. | Calculate the stellar semimajor axis $a$ and eccentricities. Then calculate the semiminor axis and finally the area. For verification, set up a Rebound simulation object. Add the two stars to the simulation, specifying their masses, positions, and velocities. Retrieve the orbital elements of one star (with the other star as the primary). Extract the semi-major axis $a$ and eccentricity $e$ from the orbit. Calculate the semi-minor axis $b$ as $b = a \times \sqrt{1 - e^2}$. Compute and return the total area of the orbit. | The formula for the total area of an ellipse: $A = \pi \times a \times b$. |
| orbital_area_star2
Determine the total area encompassed by the orbit of star2. | Same as above, but with $A = \pi \times a_2 \times b_2$. | - |
| roche_lobe_radius
Determine the Roche lobe radius of star1. | Start by calculating the stellar masses and total semimajor axis. The Roche lobe radius $R_L$ of star1 can be calculated as: $R_{L1} = \frac{0.49 \cdot q^{2/3}}{0.6 \cdot q^{2/3} + \ln(1 + q^{1/3})} \cdot a_{\text{tot}}$ where $q$ is the mass ratio of the stars. | Approximate formula from Eggleton (1983). |
| specific_angular_momentum
Determine the specific angular momentum of the system. | Find relative positions and velocities between the two stars. Calculate the specific angular momentum by computing the cross product of relative position and velocity vectors, then find its average magnitude. For verification, set up a Rebound simulation. Add both stars to the simulation with their masses, positions, and velocities. Retrieve the orbital parameters of star2 relative to star1 and extract the specific angular momentum. | The specific angular momentum is $\vec{h} = \vec{r} \times \vec{v}$ |

| | | |
|---|---|---|
| travel_time_orbital_20per_path
Calculate the time needed for star1 to travel 20% of the distance along its orbital path. | Calculate the orbital period and time of pericenter passage. Isolate the data corresponding to one complete orbit by selecting the rows between the pericenter passage and one full period later. Next, calculate the changes in the true anomaly and the radial distances and their differences. Use the formula for arc length in polar coordinates to compute the arc length for each time step: $ds = \sqrt{r^2 + \left(\frac{dr}{d\nu}\right)^2} \cdot d\nu$. Sum these arc lengths to get the cumulative path distance along the orbit. The total perimeter of the orbital path is the last value of this cumulative sum. Set a target distance corresponding to 20% of the total perimeter, and find the index where the cumulative path distance is closest to this target. Finally, return the time difference between this point and the time of pericenter passage. | - |
| virial_theorem
Determine if the Virial Theorem is satisfied in this system. Answer True if the Virial Theorem is satisfied or False if it is not. | Calculate the distance between the two stars using their 3D positions, and compute the gravitational potential energy for each time step.
Compute the velocities of both stars by taking the gradient of their positions with respect to time, and then calculate the kinetic energy for each star.
Compute the mean kinetic (K) and potential (U) energies over the entire orbit. According to the Virial Theorem, for a system in equilibrium, $2K + U = 0$. Check if the absolute value of $2K + U$ is within 0.1% of the absolute value of U. Return True if it is. | - |

## C. Choosing task-specific thresholds for budget-obs

We define task-specific thresholds to evaluate agent solutions derived from a 100 observation budget. These thresholds are based on the performance gap between our expert reference solution (using full simulation data) and the same solution but using a non-strategic observation approach with 100 uniformly sampled time points.

To establish these thresholds, we first calculate the percent difference between expert-ref-100 and expert-ref(full-obs) for each simulation-task pair. As shown in Figure 5, each black dot represents this difference for a specific pair, grouped by task. Red horizontal lines indicate our final thresholds, set to reflect typical performance for each task category.

Tasks that rely on global trends, such as determining the orbital period, are easily solved by uniform sampling and therefore receive strict thresholds. In contrast, tasks that are harder to solve with only 100 observations receive more lenient thresholds. These might require, for example, precise knowledge of local features, such as finding the maximum velocity of a star.

For particularly challenging tasks where uniform sampling fails dramatically (e.g., modified gravity exponent estimation showing >1000% error), we set a reasonable threshold (e.g. 70%) and validate that it is achievable through expert observation strategies. For instance, we validate that modified gravity exponents can be determined to within 2% accuracy through

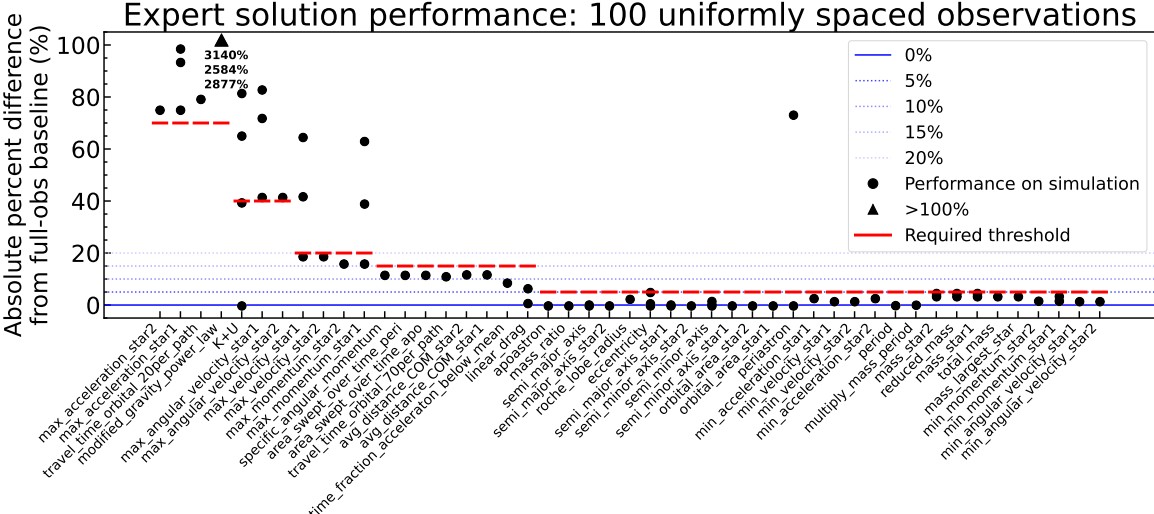

*Figure 5.* **Finding task-specific thresholds based on expert solutions performance without planning (expert-ref-100)** Each black dot represents the absolute percent difference (relative to a solution with access to the full simulated data) for simulation-task pairs grouped by task, using 100 uniformly spaced observations. Scenarios yielding large differences (e.g., 30-100% or more) show that naive uniform sampling is inadequate as a planning strategy. Red horizontal lines mark the final success threshold we choose for each *task*, that we use for all the simulations that task is based on.

strategic physics-informed sampling.

## D. Another case study on planning

One task in Gravity-Bench involves finding the periastron distance, the closest approach between two stars in their orbit. This becomes especially difficult for one of our simulation with a single highly elliptical orbit, where the stars spend less than 2% of their orbital period near this closest point (Figure 2). Our uniform sampling baseline (expert-ref-100) misses this narrow window completely, resulting in a 72% error. As shown in Figure 6, AI agents typically fixate on an early orbital phase containing a close (but non-minimal) separation, missing the true periastron that occurs later in the orbit.

Expert-level solutions with strategic planning can solve this task through initial broad sampling to identify potential regions that contain the minimum separation of stars, further analysis of the two closest regions: the start and end of the orbit, focused refinement on the late-orbit phase which shows tighter separations, and iterative sampling around the minimum. This approach achieves 2% accuracy using just 50 observations, compared to the 93% error from uniform sampling. We set a 5% success threshold for this task, reflecting both the expert solution's performance and the easier variants present in other simulations.

## E. Mass assumptions

Table 3 highlights a consistent pattern across evaluated models: solutions that shortcut the derivation and incorrectly assume stellar masses (e.g. setting both masses to 1.0) correlate strongly with incorrect answers. In every model we test, a larger fraction of incorrect responses rely on these arbitrary mass assumptions than correct ones. GPT-4o shows the largest gap, indicating it is particularly prone to incorrectly assuming the stars have equal mass or a mass equal to 1 gram or kg (what one might call a science hallucination). A small fraction of correct responses also contain mass assumptions, suggesting that occasionally these shortcuts coincidentally lead to the right solution, but this behavior avoids a principled approach to solve these problems.

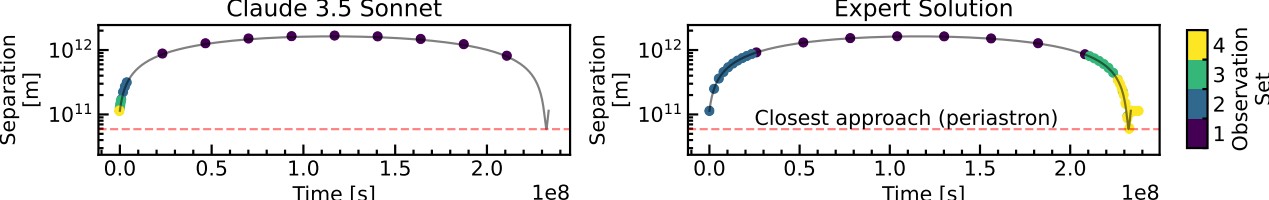

*Figure 6.* **Comparison of uniform and expert observation strategies for finding the closest approach of the stars in a highly elliptical single-orbit system.** The plots show the separation of the stars over simulation time. The left panel (Claude 3.5 Sonnet) samples at a sparse, uniform grid of times, missing the narrow periastron window near the end and yielding a distance over 93% larger than the true minimum. By contrast, the expert solution (right) first samples broadly, then zooms into the first and last parts of the orbit, then decides to continue refining observations in the last part of the orbit. This adaptive strategy locates the periastron to within 2% accuracy using the same observation budget. Circles mark observations, with brighter hues indicating later observations.

*Table 3.* **Prevalence of unjustified mass assumptions in model solutions** Analysis of 122 mass-dependent solutions shows AI models frequently assume stellar masses rather than calculating them. Percentages show how often these shortcuts appear in incorrect vs. correct solutions when using full observational data. Mass assumptions were identified by the following patterns: center-of-mass calculation "`(df['star1_x'] + df['star2_x'])/2`" and explicit mass assignments: ("`star1_mass = 1.0`", "`star2_mass = 1.0`", "`m1 = m2`", "`m1 = 1.0`", "`m2 = 1.0`", etc.). Results for other models are averages of three runs.

| | % of incorrect solutions that include a mass assumption | % of correct solutions that include a mass assumption |
|---|---|---|
| claude-3-5-sonnet-20241022 | $24.6 \pm 2.6$ (23/94) | $9.8 \pm 2.5$ (2/27) |
| claude-3-5-haiku-20241022 | $25.9 \pm 3.6$ (26/100) | $18.5 \pm 2.7$ (4/21) |
| gpt-4o-2024-11-20 | $33.0 \pm 4.8$ (32/97) | $9.3 \pm 2.7$ (2/25) |
| gpt-4o-mini-2024-07-18 | $22.1 \pm 0.6$ (22/99) | $9.0 \pm 2.6$ (2/22) |
| o4-mini-high-2025-04-16 | $28.6 \pm 6.5$ (11/39) | $12.1 \pm 0.5$ (10/82) |

# F. Prompt Templates

Prompt for `budget-obs-100`

```
You are tasked with solving the following physics problem related to a binary star
    system. You are provided observations of each star's position over time,
    (t,x,y,z), in units of seconds and meters.

### Problem Description
Determine the total energy (K + U) for the system in joules.
You must provide your answer in units of J.

### Additional Instructions
To complete this task, you have access to the following tools and data:
1. An observational tool called 'Observe' that allows you observe the system at
specific times of your choosing.
2. A code interpreter that can execute Python code.

When using 'Observe':
1. The 'times_requested' parameter should be a list that can contain any values in
    the time window [0.0, 7.21e+09] seconds. You cannot request negative times. The
    upper limit for the time window was chosen to guarantee that the problem is
    solvable with an appropriate sampling of observations using the total
    observational budget.
2. You can observe the system at any time within the time window, even if it is in
    the past compared to the last observation.
3. You can observe the system up to a total of 100 times and you can observe up to
    10 times per observational request which is the maximum length of the
    'times_requested' list.
4. After each observation, the dataframe 'row_wise_results.df' will be updated. It
    contains columns: time, star1_x, star1_y, star1_z, star2_x, star2_y, star2_z.
    You can access it using the code interpreter tool. For example, to access the
    first five rows, print(row_wise_results.df.head(n=5))

When using the code interpreter:
1. Always use print() to display results.
2. Do not use read_csv or attempt to load the DataFrame, as it is already pre-loaded
Important reminder: Repeated tool access is enabled until you have found the answer
    and have submitted it with the 'submit_answer' tool.
```

Prompt for `full-obs`

```
You are tasked with solving the following physics problem related to a binary star
    system. You are provided observations of each star's position over time,
    (t,x,y,z), in units of seconds and meters.

### Problem Description
Determine the apoastron of the system's orbit.
You must provide your answer in units of m.

### Additional Instructions
To complete this task, you have access to the following tools and data:
1. A DataFrame 'df' containing columns: time, star1_x, star1_y, star1_z, star2_x,
    star2_y, star2_z.
2. A code interpreter with 'df' pre-loaded that can execute Python code.

When using the code interpreter:
1. Always use print() to display results.
2. Do not use read_csv or attempt to load the DataFrame, as it is already pre-loaded
Important reminder: Repeated tool access is enabled until you have found the answer
    and have submitted it with the 'submit_answer' tool.
```

*Figure 7.* Exact agent prompts used in the two evaluation regimes. The first box applies when the agent is limited to a budget of 100 observations (`budget-obs-100`), the second when the agent receives the complete table of observations (`full-obs`). Two example problem descriptions are shown.

