# OpenReview forum: "Gravity-Bench-v1: A Benchmark on Gravitational Physics Discovery for Agents"
_ICML.cc/2025/Conference — ICML 2025 poster_

### Official Review · Reviewer_XGag · 2025-03-10

**Overall Recommendation:** 4

**Summary:**

This paper introduces Gravity-Bench-v1, a novel benchmark for evaluating LLM agents on physics discovery tasks. The authors design an environment that simulates gravitational dynamics with high precision, where agents must strategically plan observations, analyze data, and reason to solve various tasks. The benchmark includes standard Newtonian physics as well as out-of-distribution scenarios (e.g., modified gravity laws) to test generalization capabilities beyond memorized knowledge in LLM agents. The work provides a rigorous framework for assessing scientific discovery capabilities.

**Claims And Evidence:**

- The paper's key research motivation that current benchmarks inadequately evaluate scientific discovery capabilities in llm agents is reasonable. I suggest authors to also provide compelling evidence of the limitations in existing benchmarks, showing the memorization vs discovery gap in previous works highlighted in sec 2.

- The claim that current agents struggle with observation planning in a budget-constrained setting is convincingly demonstrated through the significant performance gap between full-observation and budget-constrained scenarios (Table 1).

- I think the claim regarding PhD-level solutions as "human reference" requires more clarification. The paper lacks detailed information about how these solutions were developed, how many human experts were involved, and what their backgrounds were. This affects the validity of using these solutions as a reference point for evaluation.

**Essential References Not Discussed:**

The issue of LLM memorization and limitation of current benchmarks for general data-driven reasoning as well as scientific discovery tasks is also discussed in some other works such as [1-3]. I would suggest authors to also consider citing these works in their related work section.

[1] Shojaee et al., LLM-SR: Scientific Equation Discovery via Programming with Large Language Models, 2024

[2] Cai et al., SciAssess: Benchmarking LLM Proficiency in Scientific Literature Analysis, 2024

[2] Xie et al., On Memorization of Large Language Models in Logical Reasoning, 2024

**Experimental Designs Or Analyses:**

The experimental design is generally sound. However, some aspects of the experimental analysis can be improved:

- The paper doesn't clearly explain why agents consistently use fewer observations than their budget allows (Figure 3). Is this a function of the prompting or an inherent limitation of the agents? How did authors explore this?

- For error analysis, it would be valuable to also see how errors and number of observations selected evolve over successive steps/iterations in agents.

- The findings about GPT-4o mini solving OOD tasks that GPT-4o fails on are intriguing but presented anecdotally rather than systematically analyzed. A more comprehensive analysis focused on these problems could be helpful to better understand the memorization vs discovery capabilities in different models.

**Methods And Evaluation Criteria:**

The metric of setting task-specific error thresholds based on the performance gap between full-observation and partial-observation PhD-level solutions is reasonable and well-justified.

However, the paper should provide more details about how agents are made aware of these budget constraints and what prompting strategies were used to encourage strategic observation planning. Also, it's worth noting the benchmark is still limited to two-body problems.

**Other Comments Or Suggestions:**

Minor Comment: I suggest adding annotation (a), (b), etc. for subplots in figures (e.g., in Figure 2 or 3) to ease referencing.

**Other Strengths And Weaknesses:**

**Strengths:**

- The benchmark addresses a gap in evaluating scientific discovery capabilities beyond memorization.

- The use of high-fidelity physics simulations provides a strong environment for testing science-focused agents.

- The inclusion of OOD cases with modified laws and simulators is particularly valuable for assessing generalization in science.

**Weaknesses:**

- The scope is currently limited to two-body gravitational problems in physics, which represents a narrow slice of scientific discovery scenarios.

- Insufficient details regarding the human reference evaluation settings, including the number of human experts involved and their evaluation protocol.

- Lack of details regarding the prompts used for different experimental cases, particularly how agents make decisions about budget allocation.

- The temporal dimension of budget utilization and error rate progression is missing from the analysis (improvement over steps, iterations in ReAct style agentic setting). Does increasing the number of iterations change the underutilization pattern observed?

- There's limited analysis of how different models approach OOD scenarios compared to other problems in current reported results.

**Questions For Authors:**

- In Table 1, why do the same models show lower computation time on full observations compared to partial observations?

- Regarding the OOD results where GPT-4o-mini solved a modified gravity task while GPT-4o did not: Have you verified this finding across multiple runs to ensure it's not anomalous? A more comprehensive analysis could provide insights about the relation between model size and discovery.

- In Section 4.6, you note that agents tend to take shortcuts rather than pursue systematic derivations. Have you analyzed whether the occurrence of these shortcuts differs between OOD cases and standard problems? Are shortcuts more or less common in novel problems?

**Relation To Broader Scientific Literature:**

The paper effectively positions itself within the landscape of benchmarks for scientific tasks and data-driven discovery.

The authors appropriately distinguish their approach and difference from existing work by highlighting the dynamic, partially observable environment and the inclusion of OOD cases to test generalization rather than memorization.

**Theoretical Claims:**

There are no formal theoretical proofs in the paper to verify.

---

> ### Author Rebuttal · Authors · 2025-04-01
>
> **(Response to 1\)** See our response to “points 2,4,5” of Reviewer f7pg.
>
> **(Response to 2\)** See our response to “point 1” of Reviewer f7pg.
>
> (**Response to 3\)** Regarding the prompting process and the manner in which the agents were informed of the budget, we adopted a minimalistic approach: the prompt merely described the observation tool, specified the budget constraints (a maximum of 100 observations, with up to 10 per request), and deliberately avoided suggesting any specific strategy. In the absence of explicit strategic guidance, it becomes increasingly evident that these AI models demonstrate an inherent tendency to prematurely converge on solutions while failing to make full use of the allocated observational budget. This behavior shows that the models often fail to approach the problem with the level of care expected in scientific problem-solving.
>
> This approach was necessary for gathering baseline results, from which we can investigate if this behavior can be mitigated via prompting or other means. The full prompt is below and will be included in the Appendix of our camera-ready version.
>
> **(Response to 4\)** We thank the reviewer for raising the point about iteratively re-calling the agent and how that would improve budget use and performance. In our current setup, agents decide when to stop and almost always do so before exhausting their observation budget. We intentionally did not force agents to use their full budget, as our goal was to assess whether they can autonomously reason about when additional data is required. We acknowledge the value of experimenting with force, i.e. increasing the number of iterations.
>
> (**Response to 5\)** We thank the reviewer for pointing this out, we agree that a deeper analysis of how different models approach OOD scenarios is important. Due to the large size of the outputs (over 1.2 million tokens per run), we were unable to perform detailed manual inspection across all cases. However, we are currently developing automated tools to analyze these responses systematically.
>
> **\*\*\*\*\*\*\*\*\*\*\*\*\*\*\*\*\*\*\*\*\*\*\*\*\*\*\*\*\*\*\*\*\*\*\*\*\*\*\*\*\*\*\*\*\*\*\*\*\*\*\*\*\*\*\*\*\*\*\*\*\*\*\*\*\*\*\*\*\*\*\*\*\*\*\*\*\*\*\*\*\*\*\*\*\*\*\*\*\*\*\*\*\*\*\*\*\*\*\*\*\*\*\*\*\*\*\*\*\*\*\*\*\*\*\*\*\*\***
> Prompt (Response to 3):
> You are tasked with solving the following physics problem related to a binary star system. You are provided observations of each star's position over time, (t,x,y,z), in units of seconds and meters.
>
> \#\#\# Problem Description
> Determine the total energy (K \+ U) for the system in joules.
> You must provide your answer in units of J.
>
> \#\#\# Additional Instructions
> To complete this task, you have access to the following tools and data:
> 1\. An observational tool called \`Observe\` that allows you observe the system at
> specific times of your choosing.
> 2\. A code interpreter that can execute Python code.
>
> When using \`Observe\`:
> 1\. The \`times\_requested\` parameter should be a list that can contain any values in the time window \[0.0, 7.21e+09\] seconds. You cannot request negative times. The upper limit for the time window was chosen to guarantee that the problem is solvable with an appropriate sampling of observations using the total observational budget.
> 2\. You can observe the system at any time within the time window, even if it is in the past compared to the last observation.
> 3\. You can observe the system up to a total of 100 times and you can observe up to 10 times per observational request which is the maximum length of the \`times\_requested\` list.
> 4\. After each observation, the dataframe \`row\_wise\_results.df\` will be updated. It contains columns: time, star1\_x, star1\_y, star1\_z, star2\_x, star2\_y, star2\_z. You can access it using the code interpreter tool. For example, to access the first five rows, print(row\_wise\_results.df.head(n=5))
> **\*\*\*\*\*\*\*\*\*\*\*\*\*\*\*\*\*\*\*\*\*\*\*\*\*\*\*\*\*\*\*\*\*\*\*\*\*\*\*\*\*\*\*\*\*\*\*\*\*\*\*\*\*\*\*\*\*\*\*\*\*\*\*\*\*\*\*\*\*\*\*\*\*\*\*\*\*\*\*\*\*\*\*\*\*\*\*\*\*\*\*\*\*\*\*\*\*\*\*\*\*\*\*\*\*\*\*\*\*\*\*\*\*\*\*\*\*\***
> Answers to direct questions:
> Q1) Partial observations (budget-obs-100) require multiple sequential tool calls, as agents iteratively request and analyze observations, compared to full-obs where all observations are available immediately in a single step.
>
> Q2) We tested each model three times. However, we acknowledge additional runs would be valuable.
>
> Q3) We have not yet specifically analyzed whether the frequency of shortcuts differs between OOD cases and standard problems.

---

### Official Review · Reviewer_f7pg · 2025-03-13

**Overall Recommendation:** 2

**Summary:**

This paper proposes a new benchmark, “Gravity-Bench-v1”, for evaluating the scientific discovery capabilities of AI agents. This benchmark is based on gravitational physics (in particular, the two-body problem), and measures the ability of agents to discover hidden physical laws in a dynamic environment. The distinctive feature of this benchmark is that the agents are given a limited observation budget (up to 100 observations), and they must plan their observations efficiently within this constraint, collect and analyze data, and solve the problem. The benchmark includes not only scenarios that follow the laws of physics in the real world, but also out-of-distribution cases (such as modified gravitational laws), to evaluate the true scientific generalization capabilities of AI. The evaluation experiments show that the latest AI models (such as o1, Claude 3.5 Sonnet, and GPT-4o) show moderate performance (up to 64%) when they have full access to data, but when their observation budgets are limited, their performance drops significantly (up to 21.5%). This suggests that current AI models have limitations in long-term planning and information gathering strategies.

## Response After Rebuttal

After carefully considering the authors' rebuttal, I maintain my evaluation as "Weak reject (2)". Here's why:

The authors provided detailed explanations regarding the PhD-level solutions, clarifying that they were verified by astrophysics experts. This point is appreciated.

However, significant limitations still remain with this benchmark. Most notably, the current version is restricted only to two-body gravitational problems. For a benchmark claiming to evaluate scientific discovery across broader domains, this scope is extremely limited. While the authors mention plans for future extensions (adding noise, 3D extension, application to other fields), the narrowness of the current benchmark restricts the contribution of this paper.

Additionally, there remains a lack of detailed analysis regarding the observation budget usage patterns. The authors state that systematic exploration is difficult due to the approximately 1.3 million tokens generated per run, but at least a detailed analysis of representative cases would have provided insights into the agents' decision-making processes.

Regarding the OOD (out-of-distribution) claims, while I understand the authors' explanation, demonstrating generalizability to broader scientific discovery tasks would require evidence beyond a single physics domain.

Indeed, the fact that even the latest AI models achieve only 40.2% under budget constraints is interesting, but while this demonstrates the difficulty of the tasks, it doesn't fully validate the value of the benchmark design itself.

Overall, while the fundamental idea behind GravityBench has merit and shows promise, the current version is too narrow in scope to recommend for acceptance at ICML. A future version implementing the extensions mentioned by the authors would likely result in a more comprehensive and valuable benchmark.

**Claims And Evidence:**

The paper's claims are generally supported by clear and convincing evidence. The main claims are the proposal of a new benchmark for evaluating AI agents' scientific discovery capabilities and the challenges that current AI models face with this benchmark. These claims are supported by comprehensive experimental results using multiple models (e.g., o1, Claude 3.5 Sonnet, GPT-4o).

In particular, the data showing the difference in performance with and without the observation budget constraint, the analysis of the agent's strategy in the observation plan, and the detailed examination of the failure mode are convincing. In addition, the evaluation in the out-of-distribution task also supports the claim that it measures the true scientific generalization ability of AI.

However, there is a lack of detailed explanation of the “PhD-level solution method”, and there is limited transparency regarding how this was actually implemented and how its accuracy was verified. In addition, the fact that the evaluation of o1 was partially limited due to budget constraints and that a comparison was not conducted under completely identical conditions for all models somewhat weakens the quality of the evidence.

**Essential References Not Discussed:**

The paper covers a wide range of related research, but there is a lack of reference to some related literature.

In particular, a more detailed comparison of specific previous research on AI systems that focus on the discovery of physical phenomena may be beneficial. For example, there is a lack of comparison with AI that performs symbolic regression of physical laws (e.g. AI Feynman, BMS) and autonomous scientific discovery systems that combine experimental design and hypothesis generation.

There are also references to the literature on partially observable Markov decision processes (POMDPs), but there is limited specific comparison with previous research on the application of POMDPs in the context of scientific discovery. A detailed discussion of the relevance of previous research in this field would make the novelty and positioning of the proposed benchmark clearer.

**Experimental Designs Or Analyses:**

The experimental design is robust and well-constructed with a clear purpose. A total of 206 task-simulation pairs were created, enabling comprehensive evaluation. The fact that each model was evaluated multiple times (3 times) under the same conditions also increases the reliability of the results.

However, there are some areas of concern. The limited number of evaluations of o1 (not evaluated under budget-obs-100) due to budget constraints makes a full comparison of all models difficult. Also, in the analysis of the observation planning capability, there is a lack of detailed analysis of why the agents do not use a large portion of the observation budget available to them. This could be useful information for a deeper understanding of the agents' behavior.

The analysis of failure modes, in particular the analysis of the correlation between the shortcut that assumes the mass of the star and incorrect answers, is very valuable. However, further verification of whether this correlation is a causal relationship would make the analysis more convincing.

**Methods And Evaluation Criteria:**

The proposed method and evaluation criteria are generally appropriate for the problem of scientific discovery. The use of Rebound, a scientific-grade physics simulation tool, provides a precise and reliable physical environment. In addition, by setting constraints on the observation budget, the decision-making process is modeled to be similar to that of actual scientific research.

The fact that the evaluation criteria include task-specific tolerance thresholds is appropriate. In particular, the fact that the thresholds are adjusted according to the difficulty of each task (e.g. 5% for easy tasks, 70% for difficult tasks) and are determined based on the performance of PhD-level solutions is reasonable.

One area for improvement is the potential to introduce noise and error into the observations, which would mimic a more realistic scientific environment. Also, the current system only handles trajectories restricted to the (x,y) plane, but it would be possible to increase the complexity and realism of the benchmark by extending it to more general 3D trajectories.

**Other Comments Or Suggestions:**

Below are some suggestions for improving the paper:

1. It would be good to add a detailed explanation of the “PhD-level solution” and increase the transparency of the implementation method and verification process.

2. Introducing noise and measurement errors into the observations would allow for a more realistic imitation of a scientific discovery scenario.

3. A detailed analysis of why the agents are not fully utilizing their observation budgets would be beneficial. This could lead to a better understanding of the agent's decision-making process and indicate directions for improvement.

4. Currently, only trajectories limited to the (x, y) plane are handled, but in the future, it is hoped that the system will be extended to more general 3D trajectories.

5. By providing more specific prospects for the possibility of extending the benchmark to a wider variety of scientific fields and phenomena, the future direction of the research will become clearer.

**Other Strengths And Weaknesses:**

The main strength of this paper is that it provides a concrete and rigorous benchmark for evaluating the complex process of scientific discovery. In particular, the design, which requires dynamic problem solving that combines observation planning and analysis, enables a realistic evaluation of the scientific capabilities of AI. In addition, the inclusion of out-of-distribution scenarios is also commendable, as it measures the true scientific reasoning capabilities of AI rather than an approach based on memorization.

One weakness is that the current version is limited to the twin-gravity problem, and it is unclear whether it can be generalized to a wider range of scientific discovery scenarios. In addition, the idealized observation conditions (lack of observation error and noise) do not fully reflect the complexity of real scientific observations.

In terms of originality, the combination of environmental benchmarking and scientific discovery evaluation is creative. In particular, the integration of scientific simulation and the concept of partial observability to create a new evaluation framework for scientific exploration is innovative.

**Questions For Authors:**

I would expect an answer to the concerns raised above, but have no additional questions otherwise.

**Relation To Broader Scientific Literature:**

This research is appropriately positioned in the context of existing efforts to utilize AI in scientific research. In particular, it discusses the relationship with various approaches, such as LLM (Galactica, etc.) that specializes in literature analysis, data-driven discovery, automated statistical modeling, and workflow automation frameworks.

The Gravity-Bench is differentiated from existing approaches in that it views discovery as a dynamic and iterative process, and it emphasizes the interaction between exploration and inference in partially observable environments. In addition, while existing benchmarks focus on rediscovering known phenomena and textbook-style problem solving, the Gravity-Bench is differentiated by including a variety of dynamic scenarios that reflect the unpredictability of real-world discovery processes.

What is particularly valuable is that this research recognizes the lack of AI benchmarks for scientific discovery and presents specific efforts to fill that gap.

**Theoretical Claims:**

This paper makes mainly experimental contributions, and contains few assertions accompanied by rigorous theoretical proofs. However, the physical foundations, such as the description of physical laws (e.g., the modified gravitational law FG ∝ r^(-2+α)) and the virial theorem (2K + U = 0), are accurately stated.

The theoretical aspects mentioned in the paper are based on established principles and laws of physics, and there is no question about their accuracy. However, the paper itself does not provide new proof of these theories, but rather applies existing physical theories to create a benchmark environment.

---

> ### Author Rebuttal · Authors · 2025-04-01
>
> We thank the reviewer for their thorough review and valuable suggestions. Below, we address the main points:
>
> **(Response to point 1)** We appreciate the feedback with respect to adding additional details regarding the human solutions. They were developed by one 2nd year PhD student, one professor, and one research scientist (PhD+10), all in astrophysics. Each one of them double checked and verified every solution. We also confirmed that the human-ref results matched Rebound’s built-in measures of the orbital parameters when available.
>
> These solutions were not developed to directly compare human and AI scores, but rather were made to serve as an empirical solution baseline to understand the difficulty of each question, which was necessary to set the threshold values an AI must exceed in order for it to get marked as “correct”. We will explicitly detail this in the revised manuscript.
>
> (**Response to 2,4,5**) We entirely agree with the recommendations for future improvements with respect to adding noise, moving to 3D, and branching to new fields. Publishing GravityBench is intentionally our first demonstration of a simulation-based scientific reasoning benchmark. We already are working on extensions including other areas of physics, like electromagnetism, and realistic observational errors. Developing these environments allow us to evaluate, and even train, AI agents on research-like tasks. Though even without these complexities, current models struggle at GravityBench, which demonstrates the benchmark's immediate value.
>
> (**Response to 3**) We agree that a deeper analysis of why agents underutilize their observation budget would be valuable. Although we began a manual analysis of agent traces, each run generated around 1.3 million tokens, making systematic exploration impractical. Therefore, we expect further analysis will require careful automation which will shed light on failure modes. The manual inspection we present suggests that agents prematurely rush toward solutions, rather than strategically and iteratively using their observation budget. We emphasize that GravityBench is self-contained in the sense that the reason for agent failures reflect limitations in reasoning and planning abilities rather than issues with the benchmark itself.
>
> **(Response to other comments)** Although we initially could not run o1 under all settings, we have now evaluated o3-mini-high, the current best-performing model, which still only achieves 40.2% under budget-obs-100. This performance demonstrates that our benchmark remains substantially challenging, as o3-mini-high still struggles with most questions, including those involving modified gravity or drag forces. These new o3-mini-high results will be provided in our camera-ready version.
>
> We appreciate the comments on systematically verifying whether the shortcut of just assuming stellar masses leads causally to incorrect answers. However, we already know that this is indeed the case. Any solution which assumes stellar masses of 1 kilogram or equal masses is physically incorrect, which directly causes these solutions to fail. This is a direct causal relationship between the assumption and the solution rather than a correlation.
>
> We acknowledge the point regarding limited references to literature on symbolic regression and AI-driven autonomous scientific discovery systems, and will expand our discussion to contextualize GravityBench with respect to these approaches. Regarding the connection to the POMDP literature, we are presently not entirely certain how "partial observability" in our benchmark relates with the standard interpretation used in idealized POMDP frameworks, as many of our problems involve regression rather than decision problems. We will clarify this relationship in the future, particularly as we introduce observational noise and uncertainty into our benchmark.

---

### Official Review · Reviewer_VSix · 2025-03-13

**Overall Recommendation:** 1

**Summary:**

The paper introduces a new benchmark called Gravity-Bench-v1 to test the discovery potential of LLMs. The which the benchmark different 2-body star systems are simulated. These simulations include out of distribution parts where the gravitationally force have a different proportion to r by adding an alpha to r^{2 + $\alpha$}. The simulation outputs start positions over time and the agents have access to them by an observation tool and a python based shell script.

**Claims And Evidence:**

The claim of the paper is to provide a benchmark to test the scientific reasoning potential of LLMs. By including out-of-distribution physics tasks it should be tested if the model is not just memorizing the solutions. However, is this really the case? And can one show this? Just thinking about an irrotational vector field in three-dimensional space. In general the inverse-square law corresponds to the property that the divergence is zero outside the source. So a general way of this law in an n-dim. Euclidean space would be that the intensity "I" of the vector field falls off with the distance "r" following the inverse (n − 1). Already here is a lot of literature about this. Furthermore, there are non-euclidean formulations even for Newtonian Cosmology (https://arxiv.org/pdf/2002.10155). These are just some examples which show that besides the simple 3-d inverse-square law a lot of other laws exist where the physics is different. It’s also easy to imagine that r^{2 + $\alpha$} can be given as an exam question in an undergraduate physics course. Also there are ton of alternatives formulations to general relativity or Modified Newtonian dynamics (MOND). Taking this beside there is another aspect: The models are closed source and are trained on data we don’t know. So how can the authors be sure that there approach is not in the data? It’s impossible to show this so making such a statement that r^{2 + $\alpha$} is out of distribution is therefore impossible to proof and then all evidence in terms of out-of-distribution is meaningless.

**Essential References Not Discussed:**

None.

**Experimental Designs Or Analyses:**

See above.

**Methods And Evaluation Criteria:**

The authors used closed source LLMs to evaluate them on out-of-distribution physics tasks. Since we don’t know the data the LLMs were trained on, the methodology does not make sense.

**Other Comments Or Suggestions:**

--

**Other Strengths And Weaknesses:**

--

**Questions For Authors:**

--

**Relation To Broader Scientific Literature:**

No really search for alternatives to Newtonian Cosmology.

**Theoretical Claims:**

None.

---

> ### Author Rebuttal · Authors · 2025-04-01
>
> We appreciate the reviewer’s detailed consideration. They raised a concern about whether our simulations overall, and the modified law of gravity in particular, are truly out-of-distribution (OOD) tests for the AI models with solutions that cannot be found online (thus incorporated into the pretraining corpus). We argue that they are indeed OOD, on the following basis:
>
> 1) A major OOD aspect to our benchmark is the methodology itself (that mimics the scientific method) the AI models need to follow to answer our problems. They need to:
>    a) Understand both the given question and data.
>    b) Plan their “observations” within the constraints of the budget. This is a physical reasoning task on its own as for example an eccentric star will spend very little time near its periastron.
>    c) Reason through a usually multi-step roadmap to the solution.
>    d) Execute the roadmap using the available tools.
> 2) The universally poor performance across the AI models (including o1) we test clearly indicates that these models are genuinely struggling with these reasoning challenges. The scores obtained (under 20% for budget-obs-100) show that our benchmark is far from saturation.
> 3) Unlike other benchmarks that rely on or mimic questions largely obtained from sources available on the internet, we specifically designed our own simulations entirely from scratch. It is highly unlikely that similar problems are available in online or offline sources, ensuring memorization from any source remains very unlikely.
>
> These points above emphasize that our modified gravity law problem is OOD. Since it is not only that an identical question is unlikely to be found in any other source, the methodology to solve it (all the steps of point 1\) makes it truly unique.
>
> To conclude, even aside from the scenarios which we label 'out-of-distribution' (those with modified gravity and drag forces), our benchmark substantially contributes by evaluating dynamic planning and iterative reasoning under limited observational budgets, critical capabilities for scientific discovery yet rarely tested by existing benchmarks. This value is independent of whether one fully accepts our label of 'out-of-distribution' for the two scenarios where we go beyond the standard gravitational force.

---

### Decision · Program_Chairs · 2025-05-01

**Decision:**

Accept (poster)

**Comment:**

This paper introduces a benchmark to test the scientific discovery abilities of LLMs by allowing them to make a limited number of observations in order to accurately predict the behavior of a 2 body star system simulation. The benchmark includes variations with modified laws of physics with the aim of being more out of distribution with respect to LLM pre-training data. The empirical results show that current SOTA LLMs struggle to plan to gather information effectively.

While the benchmark is limited to 2-body simulations, the current empirical evaluations demonstrate a limitation of the current models, and it is not in-general unreasonable that broader capabilities are first measured on a restricted domain. However, the authors should make sure that it is clear from the framing of the paper that this currently aims to measure this restricted subset of the general capability. Moreover, the authors should be sure the justification of all claims are made clear, such as those about "PhD-level" solutions.